# Circulating Biomarkers for the Early Diagnosis and Management of Hepatocellular Carcinoma with Potential Application in Resource-Limited Settings

**DOI:** 10.3390/diagnostics13040676

**Published:** 2023-02-11

**Authors:** Annabelle Pan, Thai N. Truong, Ying-Hsiu Su, Doan Y Dao

**Affiliations:** 1School of Medicine, Johns Hopkins University, Baltimore, MD 21205, USA; 2Department of Internal Medicine, Campus in Thanh Hoa, Hanoi Medical University, Thanh Hoa 40000, Vietnam; 3Department of Translational Medical Science, The Baruch S. Blumberg Institute, Doylestown, PA 18902, USA; 4Center of Excellence for Liver Disease in Vietnam, Johns Hopkins University of Medicine, Baltimore, MD 21205, USA

**Keywords:** hepatocellular carcinoma, liver cancer, early detection, circulating biomarkers, resource-limited settings, low-to-middle-income countries (LMICs)

## Abstract

Hepatocellular carcinoma (HCC) is among the world’s third most lethal cancers. In resource-limited settings (RLS), up to 70% of HCCs are diagnosed with limited curative treatments at an advanced symptomatic stage. Even when HCC is detected early and resection surgery is offered, the post-operative recurrence rate after resection exceeds 70% in five years, of which about 50% occur within two years of surgery. There are no specific biomarkers addressing the surveillance of HCC recurrence due to the limited sensitivity of the available methods. The primary goal in the early diagnosis and management of HCC is to cure disease and improve survival, respectively. Circulating biomarkers can be used as screening, diagnostic, prognostic, and predictive biomarkers to achieve the primary goal of HCC. In this review, we highlighted key circulating blood- or urine-based HCC biomarkers and considered their potential applications in resource-limited settings, where the unmet medical needs of HCC are disproportionately highly significant.

## 1. Introduction

Hepatocellular carcinoma (HCC) is the fourth most commonly diagnosed cancer worldwide and the third leading cause of cancer-related death [1]. The burden of HCC is particularly concentrated in Eastern and South-Eastern Asia (Mongolia, Cambodia, Vietnam); and Northern and Western Africa (Egypt, the Gambia, Guinea) [2,3,4]. Most HCC cases occur secondary to cirrhosis caused by chronic viral hepatitis, with chronic hepatitis B virus (HBV) being the most prominent disease etiology, followed by alcohol, chronic hepatitis C virus (HCV), and then other causes, such as non-alcoholic fatty liver disease and aflatoxin exposure [2].

Several studies have found that the 5-year survival rate of HCC increases substantially when the disease is caught early, prior to the cancer infiltration and metastasis [5,6,7]. Given the strong association between early diagnosis and survival [8], there are many efforts towards increasing the accuracy and accessibility of early detection measures, particularly in high-risk populations, such as those with advanced fibrosis or cirrhosis, secondary to chronic HBV, HCV, HIV, or other chronic liver diseases [9].

Guideline-concordant recommendations for HCC surveillance should be offered to all patients with cirrhosis, or chronic hepatitis B carriers with a family history of HCC, Asian-born males aged 40 years or older, Asian females aged 50 years or older, or African-born individuals aged 20 years or older [9,10].

To date, the most widely used screening test for HCC is alpha-fetoprotein (AFP), combined with the right upper quadrant abdominal ultrasound [11]. However, these methods are limited by low sensitivity (40–60%) [12], variable specificity with AFP, and operator dependent accuracy of the ultrasound. To improve these screening tools, much research has been done to elucidate HCC biomarkers, defined by the US National Cancer Institute as “any substance, structure, or process that can be measured in the body or its products, and influence or predict the incidence of outcome or disease” [13].

The ideal biomarker for cancers should be noninvasive, accurate, and straightforward to interpret [14]. For a biomarker to be applicable in widespread screening in resource-limited settings, additional considerations include affordability, ease of use, and portability for point-of-care testing [15]. This review will discuss the current landscape of circulating biomarkers that show promise in their suitability for LRS (Figure 1). For this review, we focused on two major sources of circulating blood and urine biomarkers.

## 2. Blood-Based Biomarkers of HCC

In recent years, there has been a substantial focus on blood-based biomarkers, which provide tumor-related information from blood. Key biomarkers related to HCC in the blood include circulating tumor cells, circulating tumor DNA and RNA, and proteins [16]. A summary of the findings is shown in Table 1.

**Table 1 diagnostics-13-00676-t001:** Clinical studies of novel, low-cost methods for detecting HCC biomarkers in the blood *.

Name of Biomarker	Tools for Diagnosis	Study Characteristics			Suitability for Resource-Limited Settings (RLS)	
		Population	Sensitivity	Specificity	Pros	Cons
CTCs	Microfluidics [16]	HCC: 14, non-HCC malignant tumor: 7 HCON: 6	85.7%	100% for HCC vs. HCON; 46% for HCC vs. HCON + non-HCC tumor	Rapid (time to result < 90 min), low cost, and user-friendly operation; Detects CTCs with high Se, and has good Sp in distinguishing no cancer vs. any cancer	Not specific for HCC vs. non-HCC cancers
	Dual-targeting functionalized, reduced graphene oxide film (DTFGF) [17]	HCC: 8 ICC: 1 HCON: 1	100%	100%	Rapid, low-cost (inexpensive reagents), potentially high accuracy	Limited testing sample size
	Imaging flow cytometry (IFC) [18,19]	HCC: 52 CCA: 05 n-ca: 12 HCON:12	85.19%	78.35%	Low cost (avoids the need for antibodies), simple	Limited testing sample size
cfDNA	DNA fragmentomics [20]	HCC: 159 ICC: 26 HCC + ICC: 7 HBV/CIR: 51 HCON: 113	96.8%	98.8%	High accuracy, low cost (using low-coverage genome sequencing), user friendly (automated)	Not specific for HCC vs. non-HCC cancers, possible over-fitting of machine learning model
ctDNA	Methylation marker panel [21]	HCC: 383 HCON: 275	83.30%	90.50%	Low cost (targeted, low-coverage sequencing);	Lack of liver disease controls
	5-hmc Seal [22]	Early HCC: 220 HBV/CIR: 129	82.70%	67.40%	Early diagnosis (avoiding more advanced treatment needed at later diagnosis), low cost (targeted sequencing)	The 5-hmC-Seal technique is patented and may be costly to use
RNA	Hyperspectral imaging [23]	HCC: 36, HCV: 4 (3 CIR) Benign liver lesions: 2 HCON: 4	>98%	>98%	Low cost, ease of use, portability, potentially high accuracy	Limited testing sample size

Abbreviations: HCC = hepatocellular carcinoma; ICC = intrahepatic cholangiocarcinoma; CCA = cholangiocarcinoma; HCON = healthy control cohort; CIR = cirrhosis cohort; CLD = chronic liver disease; HBV = hepatitis B virus; HCV = hepatitis C virus; n-ca = noncancerous liver disease, Se = sensitivity; Sp = specificity. * This table excludes techniques that have not been tested for diagnostic accuracy; these are listed in Table 2.

### 2.1. Nucleic Acid Biomarkers

#### 2.1.1. Circulating Tumor Cells (CTCs)

Recent studies have shown that the diagnosis of HCC is correlated with the quantity of CTCs in patient blood [24]. However, as most CTCs are destroyed by the host immune system, absolute CTC counts are often lower than 10 cells per mL of blood or plasma [16], and a highly sensitive and simple identification is necessary for it to be suitable for HCC diagnosis in LRS. For instance, a study published by Wang and coworkers in 2021 describes a 3D-printed microfluidic chip that can capture and quantify CTCs within 90 min under optimal conditions [16]. In general, microfluidics function by exploiting the physico-chemical characteristics of two immiscible fluids within microfluidic channels, which allows the creation of microdroplets that serve as individual reaction chambers to contain and detect as few single cells or molecules as possible. The microfluidic chip developed by Wang et al. for detecting CTC includes a 3D-printed platform for introducing reagents into the chip, allowing a fully automated process that is simple to use and reduces antibody consumption by 90%, significantly reducing cost. Under stable temperature and pressure conditions, the chip detected CTCs in 12 out of 14 individuals with clinically diagnosed HCC, and zero out of six healthy controls. An important limitation of this method is that it does not distinguish between different cancers. However, the ability to detect small quantities of CTCs with high sensitivity, low cost, and user-friendly operation can still be valuable for screening populations at a high risk of developing HCC.

A separate study detecting HCC CTCs, specifically, was described by Wu and colleagues in 2019. This method, named dual-targeting functionalized graphene film (DTFGF), involves attaching two compounds to a graphene oxide film: antibodies targeting epithelial cell adhesion molecules (EpCAM), and nanoparticles targeting HCC cell-specific asialoglycoprotein receptors (ASPGR). When HCC CTCs are captured by anti-EpCAM antibodies, and subsequently endocytose nanoparticles specific to ASPGR, this leads to rhodamine fluorescence that can be detected through simple fluorescent microscopy. When tested with patients’ blood, the DTFGF could detect HCC CTCs, with increasing fluorescence in patients with clinical diagnoses of HCC stages III, IVB, and IV. No HCC CTCs were detected in two controls: one with intrahepatic cholangiocarcinoma and one healthy volunteer. While this study involved a very small sample size, its results suggest a promising direction for future research into an affordable and simple method for detecting HCC CTCs [17].

To entirely circumvent the need for expensive antibodies, some studies detecting CTCs have also utilized simple imaging flow cytometry (IFC). In 2016, Liu and coworkers analyzed 81 subjects, including 52 with HCC, five with cholangiocarcinoma, 12 with non-cancerous liver disease, and 12 healthy controls, and determined the sensitivity of IFC to be 85%, with a specificity of 78% [18,25].

#### 2.1.2. Cell-Free DNA (cfDNA)

Cell-free DNA (cfDNA), defined as circulating fragmented DNA that exists in both healthy and diseased individuals, has shown promise for diagnosing HCC [26]. Two major approaches to extracting cancer-specific information from cfDNA are next-generation sequencing (NGS) and targeted PCR [26]. NGS can be used to construct “fragmentomic” profiles from individual blood samples, which can then be run through a logistic regression or machine learning, in order to classify samples into various disease states. After collection and purification, plasma samples can be sequenced to varying depths, with higher depths producing longer sequences and a higher coverage. The sequencing data are then cleaned and analyzed, and classified as HCC or non-HCC, using either machine learning or simpler statistical means, such as a logistic regression.

To address the high cost of sequencing, it is possible to use machine learning to augment the classification accuracy from the low-depth sequencing [27]. In 2021 Zhang et al. tested this technique on the detection of HCC [20]. Using an input of shallow whole-genome sequencing (as low as 1x coverage), their machine learning model was trained on a cohort of 362 participants, of which 192 patients had liver cancer (159 HCC, 26 intrahepatic cholangiocarcinomas (ICCs), and seven combined HCC-ICC (cHCC-ICC)), and 170 were non-cancer controls (of which 53 had liver cirrhosis or chronic hepatitis B). The model was then tested on a separate cohort of 354 participants, of which 189 had liver cancer (157 HCC, 26 ICC, and six cHCC-ICC) and 165 non-cancer controls (of which 50 had liver cirrhosis or chronic hepatitis B). In the complete test cohort, their model could distinguish cancer from non-cancer with a sensitivity of 96.8% and specificity of 98.8% (AUC 0.995). Within the cancer patients alone, the model could distinguish HCC from non-HCC, with an AUC of 0.77.

A key limitation of this study is the possibility of overfitting the statistical model, although several precautions were taken to prevent overfitting. The model is also limited in distinguishing HCC from non-HCC; however, it shows promise for distinguishing HCC from non-cancerous liver disease, which may have a great clinical utility.

This whole-genome cell-free DNA fragmentome analyses have recently been demonstrated as promising in an international cohort from the US, EU, or Hong Kong. Foda Z et al. showed that among the 724 individuals with HCC (*n* = 165), or high risk or average risk (*n* = 234), a machine learning model that incorporated multi-feature fragmentome data had a sensitivity for detecting cancer of 88% in an average-risk group at 98% specificity, and 85% among the high-risk population at 80% specificity. The performance of the model was subsequently validated in an independent cohort (*n* = 223). Importantly, the authors also demonstrated that cfDNA fragmentation changes reflected genomic and chromatin changes in HCC, including from transcription factor binding sites [28].

The greatest limitation to sequencing, when applied to low-resource settings, is that even low-coverage sequencing can still be expensive, particularly compared to single nucleotide polymorphism (SNP) array genotyping or assays for individual proteins [29]. Furthermore, any method involving cfDNA extraction requires using a centrifuge at −80 °C storage, and specialized storage tubes that preserve the quality and purity of cfDNA [26]. However, sequencing technology has become more and more efficient over time, and the authors believe these approaches are worth mentioning because of their high accuracy and likelihood of affordability in the near future.

#### 2.1.3. ctDNA

In cancer patients, circulating tumor DNA (ctDNA) comprises a small proportion (<1%) of the total cfDNA [30]. This portion can be specifically identified using known primers of tumor-specific DNA. The dominant strategies in the field of ctDNA detection can be generally grouped into DNA mutation, DNA methylation, and 5hmC modification [31].

##### DNA Mutations

Dozens of genes have been identified as differentially expressed during HCC [32]. When analyzed as individual mutations, the range of sensitivity ranges from 9 to 86%, with a tradeoff of decreased specificity at higher sensitivities. Electrochemical biosensors are a promising tool for detecting the single-nucleotide polymorphism (SNP) subset of DNA mutations in low-resource settings. Biosensors can avoid the need for thermal cycling and other costly techniques and be more user friendly and portable for the point-of-care use [33]. One such biosensor, designed by Huang and coworkers, tested a nested hybridization chain reaction on detecting PIK3CA E545K ctDNA in the serum of 23 breast cancer patients, and pleural effusion samples from 25 HCC patients. The sensor could detect this form of ctDNA in six breast cancer serum samples and two HCC pleural effusion samples, which demonstrates promise for its clinical applicability [34]. Such sensors should theoretically scale well with the addition of multiple molecular targets, which could improve their diagnostic accuracy and clinical utility. However, as noted by Li and coworkers, there is presently a disconnect between laboratory prototypes and clinical devices, which is at odds with the advantages of these devices reported by laboratories [33].

##### DNA Methylation

DNA methylation is an epigenetic modification that regulates gene transcription [35]. Changes in DNA methylation are highly implicated in carcinogenesis, particularly through hypermethylation of tumor suppressor genes and hypomethylation in the oncogenes [36]. To develop a cost-effective means of analyzing DNA methylation levels, Xu and colleagues have developed a combined diagnostic score, produced through machine learning, and using data based on the methylation level of eight marker genes. When their low-cost panel of DNA methylation was tested on a sample of 383 HCC tumor samples and 275 normal healthy tissue samples, the combined diagnostic score could distinguish HCC from healthy tissue with an AUC of 0.944 [21].

##### 5-Hydroxymethylcytosine

Epigenetic markers 5-hydroxymethylcytosines (5hmCs) are generated from the oxidation of 5-methylcytosines by ten to eleven translocation enzymes. The presence of 5-hmC generally reflects gene expression activation, and the detection of decreased 5-hmC has been shown to be useful for detecting cancer pathobiology [37]. Similar to DNA methylation, a cost-effective approach for measuring 5-hmC involves a technique that avoids deep sequencing [22]. Cai and coworkers developed a method called the “5hmC-Seal”, that characterizes genome-wide 5hmC profiles in cfDNA and only requires sequencing of enriched 5hmC-containing cfDNA fragments at low coverage. The 5-hmC-Seal was performed with a 32 gene diagnostic model on 2554 subjects, of which 1204 had HCC, 392 had chronic hepatitis B or liver cirrhosis, and 958 were healthy individuals. The resulting AUC in a validation set was 0.884.

#### 2.1.4. RNA

RNA transcriptome analysis can be applied to distinguish HCC samples from non-HCC samples and reveal RNA segments that are differentially expressed in the HCC [38]. However, conventional methods for transcriptome analysis, such as qtPCR or next-generation sequencing (NGS), are prohibitively costly and time-consuming.

A novel and affordable alternative to sequencing is hyperspectral imaging (HSI). HSI is an imaging technique that collects a “spectral signature” of a substance, based on its reflection, transmission, and absorption of electromagnetic radiation at various wavelengths. HSI has previously been used in a variety of fields, including astrology, archeology, and forensic medicine, showing great success in discerning molecular differences between samples [39]. In biomedicine, HSI has been used for various cancer detections, both from tissue and blood samples [40,41,42]. In 2021, Aboughaleb and coworkers tested hyperspectral imaging on RNA purified from the serum of 36 HCC patients, 24 healthy controls, four people with chronic hepatitis C (among which three had liver cirrhosis), and two people with benign liver lesions [23]. After extracting and purifying RNA, HSI was performed using a cheap commercial laser pointer and a mobile CCD camera. They identified an optimal wavelength for distinguishing samples to be within the red band (633–700 mm), which agreed with a previous study that considered the spectral signature of the liver tumor versus normal tissue [43]. By analyzing visual differences in the spectral signatures, they successfully distinguished every sample of HCC from non-HCC [23].

**Table 2 diagnostics-13-00676-t002:** Low-cost techniques for detecting HCC proteins in the blood.

Name of Biomarker	Tools for Diagnosis	Accuracy	Standard Assay for Accuracy Determination	Suitability for Resource-Limited Settings (RLS)	
				Advantages	Disadvantages
AFP	Quantum dot [44]	Sensitivity: 95.62% Specificity: 96.08	Commercial electrochemi-luminescence immunoassay	Low cost, rapid (<10 min), requires only 50 μL of sample serum.	Limited sensitivity of AFP
	Microfluidics [45]	r^2^ = 0.9812	Roche electrochemi-luminescence immunoassay	Low cost, rapid (40 min), easy to operate, requires only 17 μL of sample serum	Limited sensitivity of AFP
AFP-L3	Microfluidics [46]	r^2^ = 0.981	LiBASys assay for AFP–L3%	Automation, low cost	Limited sensitivity of AFP-L3

Note: these are separated from non-protein techniques, because they were not measured accurately based on their ability to diagnose HCC; rather, they are compared against standard assays for these biomarkers.

### 2.2. Protein Biomarkers

#### 2.2.1. Alpha-Fetoprotein (AFP)

AFP is a tumor-associated protein that is the most widely used biomarker for HCC worldwide [47]. Under normal conditions, AFP is a major fetal glycoprotein, synthesized in utero by the embryonic liver, and presents at low levels in adults [48]. AFP can be secreted by HCC cells, which can suggest cell maturation arrest in a pseudo-embryonic state [48]. Using elevated AFP to diagnose HCC is controversial largely due to low sensitivity and specificity, particularly AFP expression is affected by patients with pre-existing liver diseases, such as cirrhosis and chronic hepatitis B virus [49,50].

In a recent systematic review and meta-analysis of studies using AFP for HCC diagnosis, Zhang and coworkers found that the optimal threshold above which an AFP test should be considered positive is 400 ng/mL. This was determined by comparing the AUC and SROC of tests that used thresholds of 20–100 ng/mL, 200 ng/mL, and 400 ng/mL [51]. At a threshold of 400 ng/mL, they found a pooled sensitivity and specificity of 0.32 (95% CI 0.31–0.34) and 0.99 (95% CI 0.98–0.99), respectively. This study captures the tradeoffs between raising the AFP threshold and decreasing the sensitivity, while increasing the specificity of test results. However, despite the limited sensitivity of AFP testing with 400 ng/mL, the AUC was determined to be 0.94, which shows good accuracy for HCC diagnosis [51].

Because of its specificity, long history of use, and affordability, AFP remains the most used and an integral component of HCC detection globally, and many novel strategies for refining HCC detection continue to retain a measurement of AFP levels [52]. Furthermore, AFP is centrally important because it is the only biomarker that has passed through all phases of biomarker development, as originally proposed by Pepe and coworkers [53]. Traditional means of detecting AFP often involve immunoassays, such as ELISA and RIA. New technologies which aim to reduce time, labor, and cost may be promising for use in low-resource settings.

##### Quantum Dot for Measuring AFP

Yang and coworkers used AFP to demonstrate the accuracy of a portable biosensor that integrates quantum dots with an immunochromatography test strip (QD-based ICTS) [44]. By combining the powerful luminescence and photostability of QDs with a stable sandwich immunoreaction, this biosensor was able to detect levels of AFP as low as 1 ng/mL in 10 min, using 50 µL of human serum. When tested on 1000 human serum samples in comparison with a standard commercial ELISA-based test kit, the sensitivity and specificity of QD-based ICTS both exceeded 95%. Given the biosensor’s rapid time-to-result, accuracy, and low cost, this technique could be useful for point-of-care testing in low-income settings [44].

##### Microfluidics for Measuring AFP

Several studies have tested the use of microfluidics for detecting AFP [54]. These utilize a variety of elements and antibodies for the adhesion and detection of their target molecule, all of which are contained in a microfluidic system constructed from paper or polymers, such as PMMA and PDMS. This technology is known to be highly affordable and suitable for point-of-care testing [54]. While some microfluidic-based biosensors use complex detection methods and expensive external equipment that is often inaccessible in low-resource settings, simple colorimetric, highly cost-effective analysis has been shown to detect AFP at concentrations as low as 1.7 pg/mL [55]. To target the user-friendliness of colorimetric detection systems, smartphone classification systems have successfully distinguished AFP-spiked serum from controls, showing promise for a simple and low-cost approach for point-of-care testing [56]. Advances in microfluidic technology are further reducing the time and reagents required to detect molecules such as AFP, which shows promise for application in low-resource settings [57].

The most important limitation of AFP use in HCC is that approximately 50% of HCCs do not secrete AFP (or AFPnegative HCC). AFP negative is defined as AFP < 20 ng/mL [58]. Thus, research to discover biomarkers for AFP negative HCC is an area of active investigation.

#### 2.2.2. Proteins Included in the GALAD Score

Some regions, particularly Japan, practice using proteins in addition to AFP to compile a GALAD score, combining gender, age, AFP, AFP-L3%, and DCP for diagnosis and surveillance [31]. DCP, also known as protein induced by vitamin K absence or antagonist-II (PIVKA-II), is an immature form of prothrombin. Elevated DCP values (≥7.5 ng/mL) have been shown to be associated with a 5-fold increased risk of developing HCC, and on this basis, DCP has received Food and Drug Administration (FDA) approval for risk assessment. AFP-L3%, a glycoprotein normally produced by fetal liver, is one of three AFP glycoforms that can be separated based on its lectin-binding characteristics, most readily with Lens culinaris agglutinin (LCA). In adults, an increase in AFP-L3% appears more specific for HCC than total AFP. It is usually presented as a percentage of the total AFP with a reference range of <10%.

A statistical model (known as ‘GALAD’: gender, age, AFP-L3%, AFP, and DCP) formally combines these three serum biomarkers together with age and gender to produce an algorithm with a better performance than its individual constituents. The GALAD model is of the form:Z = −10.08 + 0.09 × age + 1.67 × sex * + 2.34 log (AFP) + 0.04 × AFP-l3% + 1.33 × log (DCP)
* sex = 1 for males, 0 for females [59].

This score has been internally and externally validated [60], and recently received a breakthrough designation from the FDA. The performance of GALAD has been evaluated as a surveillance test for HCC in the US, UK, Germany, Japan, and Hong Kong in case-control studies, but has not yet been evaluated in low-to-middle-income countries (LMICs) [60]. Furthermore, GALAD has yet to undergo phase IV (prospective cohort) biomarker studies for the clinical surveillance of HCC early detection.

Furthermore, the clinical applicability of the GALAD score to low-resource settings may be limited due to the raised cost of adding additional biomarkers. The potentially increased sensitivity must be balanced with the possibility of higher false positives and the additional cost of additional biomarkers [61].

##### Microfluidics for Analyzing AFP-L3%

In 2009, Kagebayashi and coworkers published a study considering the use of microfluidic devices to detect AFP-L3% concentration. Their system was tested on spiked solutions of AFP-L3%, and had a highly consistent limit of detection of 0.1 ng/mL (clinically relevant concentrations are usually in excess of 10% of AFP = 20 ng/mL, or 2 ng/mL [46]). Their system offers the advantages of low running cost, automation, and a high sensitivity.

##### DCP

Several studies have found that des-carboxy-prothrombin (DCP), also known as protein induced by vitamin K absence of antagonist-II (PIVKA-II), is an effective biomarker for HCC [62]. Some clinical findings have suggested that DCP/PIVKA-II should not be used alone [63]; however, many studies suggest that it improves diagnostic accuracy when combined with AFP and AFP-L3% [64,65,66], particularly in distinguishing HCC patients from those with cirrhosis [67] and NAFLD (non-alcoholic fatty liver disease) [68], the latter which is a fast-growing cause of HCC.

In 2019, Huang and coworkers reported developing a portable biosensor to detect PIVKA-II, that uses a test card containing up-converting phosphor technology [69]. This biosensor was tested on 498 serum samples from 228 patients with HCC, 170 with benign liver lesions, and 100 healthy controls. It could distinguish HCC from non-HCC with an AUC of 0.85 [69]. For patients with HCC early detection, DCP had an AUROC of 0.72 [70].

Clinically, judgement in using DCP is needed, as there is a lack of formal phase III or IV biomarker validation studies, that are considered late stages, with prospective, longitudinal, evidence-based performance ready for clinical use.

## 3. Urine-Based Circulating Biomarkers

Urine contains an abundance of DNAs, RNAs, proteins, circulating tumor cells, exosomes, and other small molecules, which can be detected as biomarkers for HCC screening, diagnosis and management (Table 3) [14]. Specifically, the biomarker needs to be small enough, approximately less than 20 kDa in atomic weight, and a correct ionic charge, to be filtered by the renal glomerulus and not re-absorbed by the tubules. Second, the marker should be specific to the cancer in question and not secondary to the effects of cancer on general physiology. Finally, the marker should be secreted in adequate amounts for accurate, repeatable detection in early disease [50]. Therefore, urine has become one of the most attractive biofluids in clinical practice, due to its easy collection approach, availability in large quantities, and noninvasiveness [71], which would increase patient acceptability and compliance [72].

Urine can be used for widespread screening and surveillance through a simple dipstick test, suited to the developing world, where cost and access to other techniques or assays can be more problematic.

Despite the strengths, urine-based biomarkers also have limitations. Urine-based biomarkers for liver cancer can be challenging to isolate, with a high confidence of specificity. Its titer is not only elevated in cancer, but can also be raised in other medical conditions, including pregnancy, trauma, or inflammation [73]. Additionally, other substances in urine may interfere with ucfDNA-based test accuracy [74]. Dietary intake and urine collection and urine cell-free DNA (ucfDNA) isolation procedures can also impact the urine test integrity. As a result, studies should carefully consider these elements during planning and implementation [14].

Furthermore, high-sensitivity biomarkers in the urine may still be expensive and time-consuming to assay, and require specialized knowledge regarding biochemistry and bioinformatics. Many efforts are being made to simplify the workflow and improve the sensitivities of these methods, and as such, many lower-cost methods are being developed. Overall, analyses of specific biomarkers in the urine may provide clinically relevant information and could be alternatives to more traditional approaches, or could complement the mutation analysis [72].

There are three types of urinary biomarkers: DNA, RNA, and protein-based/metabolites, and we discussed in the review [75,76].

**Table 3 diagnostics-13-00676-t003:** Summary of urine-based biomarkers.

Ref. #	Study Population	Methods/Platform	Country of Origin of Patient Samples	Significantly Changed Metabolites or Pathways
[77]	HCC: 21 HCON: 24	RPLC–qTOF–MS, HILIC–qTOF–MS	China	Arginine and proline metabolism (creatinine), alanine and aspartate metabolism (carnitine, acetyl-carnitine), fatty acid oxidation (several acylcarnitines), nicotinate and nicotinamide metabolism (N-methyl nicotinic acid), phenylalanine metabolism (phenylacetylglutamine), purine metabolism (hypoxanthine) and lysine degradation (carnitine)
[78]	HCC: 33 CIR: 21 HCON: 26	LC-QTRAP MS	China	Nucleosides, bile acids, citric acid, several amino acids, cyclic adenosine monophosphate, glutamine, and short- and medium-chain acylcarnitines; Purine, energy, and amino acid metabolism
[79]	HCC: 20 HCON: 20	GC/MS	China	Glycine, hypoxanthine, xylitol; Glycine and xylitol metabolism
[80]	HCC: 82 HCON: 71 Benign liver tumor: 24	GC-TOF MS + UPLC-qTOF MS	China	Bile acids, histidine, and inosine; Bile acids, free fatty acids, glycolysis, urea cycle, and methionine metabolism
[81]	HCC: 16 CIR: 14 HCON: 17	^1^H-NMR	Egypt	Glycine, trimethylamine-N-oxide, hippurate, citrate, creatinine, creatine, and carnitine
[82]	HCC: 42 CIR: 47 CHB: 46 (n-cir) HCON: 7	^1^H-NMR	Bangladesh	Acetate, creatine, creatinine, dimethylamine, formate, glycine, hippurate, and trimethylamine-N oxide
[83]	HCC: 25 HCON: 12	UPLC–qTOF–HDMS	Chinese	Glycocholic acid expression
[84]	HCC: 25 HCON: 12	LC-qTOF-MS	Chinese	Bile acid biosynthesis, citric acid cycle, tryptophan metabolism and urea cycle metabolism
[85]	HCC: 55 CHV: 40 HCON: 45	GC-MS	Egypt	Glycine, serine, threonine, proline, and citric acid: higher; Urea, phosphate, pyrimidine, arabinose, xylitol, hippuric acid, xylonic acid and glycerol: lower
[86]	HCC: 13 CLD: 35	^1^H-NMR	United Kingdom	Carnitine and formate: higher; Citrate doublet, hippurate, p-cresol sulfate, creatinine methyl and creatinine methylene: lower

Abbreviations: HCC = hepatocellular carcinoma; HCON = healthy control cohort; CIR = cirrhosis cohort; CLD = chronic liver disease; CHB = Chronic hepatitis B; CHC = Chronic hepatitis C; n-cir = non-cirrhosis; GC = gas chromatography; MS = mass spectrometry; LC = liquid chromatography; QTRAP = hybrid triple quadrupole linear anion trap; HDMS = high-definition mass spectrometry; HILIC = hydrophobic interaction chromatography; NMR = nuclear magnetic resonance; qTOF = quadrupole time of flight; RPLC = reversed phase liquid chromatography; TOF = time of flight; TQ = triple quadrupole; UPLC = ultra-performance liquid chromatography.

### 3.1. Urine Cell-Free DNA (ucfDNA)

Apoptosis occurs in cancer cell turnover and is thought to be the primary source of circulation of cell-free DNAs [87]. Some of these cfDNAs get filtered into urine and are subsequently isolated and detected [88]. From the technical standpoint, when these urinary cfDNAs pass through glomerular filtration, the integrity of this ucfDNA is an important consideration [87].

The discovery by Lin et al. highlighted the proof of the concept of isolating and detecting p53 mutations (codon 249) in the urine of patients with HCC. The authors showed an excellent concordance of urine TP53 and circulating TP53, and subsequently demonstrated the detection of urine TP53 in 50% of the patients with HCC [89].

Next, in 2014, Su et al. demonstrated that multiple noninvasive biomarkers, including both genetic markers (mutations of TP53 249 T) and epigenetic markers (aberrant methylation of RASSF1A (mRASSF1A) and GSTP1 (mGSTP1) genes), were detected in in the urine of patients with HCC. Uniquely, these mutation biomarkers are derived from different cancer pathways, which may represent the heterogeneity inherence to HCC. Ultimately, three markers (TP53 249 T, mRASSF1A, and mGSTP1) were chosen for development in HCC screening [90].

In 2022, the same investigative team further established the potential clinical utility of urine ctDNA in a case-control study with prospective collection of urine [91]. In this study, 609 participants were enrolled, comprising 186 early-stage HCC cases, 144 cirrhosis, and 279 chronic hepatitis B carriers without cirrhosis or HCC. The urine ctDNA test with AFP accurately discriminated HCC from non-HCC, at 80% sensitivity and 90% specificity. The urine ctDNA test, when combined with AFP, outperformed either the urine ctDNA test alone or AFP alone, with the AUCs of 0.854, 0.744, and 0.912, respectively, [91]. In other words, the urine ctDNA test was able to detect 30% more HCC cases compared to AFP alone. The urine ctDNA test detected 49% of all “AFP-negative” HCC cases.

Table 4 summarizes the sensitivity and specificity of urine ctDNA test compared to other commonly used tests.

To facilitate the implementation of the HCC urine ctDNA test more readily into clinical use, urine ctDNA stability has been engineered (EDTA preservative) and demonstrated to remain consistent between room temperature vs. frozen temperature (−20 °C) for convenience and cost. The detection of HCC-associated urine DNA markers was analyzed in samples stored at RT for 7 days in EDTA preservative versus samples stored and transported at RT to facilitate low cost and use of the test kit in RLS.

### 3.2. Urine Protein and Metabolite Biomarkers

Unlike the serum, urine does not normally contain large amounts of protein: healthy individuals secrete < 150 mg/24 h. The glomerular proteinuria pattern can be identified by the presence of albumin, and substantially lower the amounts of β-globulin alone (selective glomerular) or α1, α2, and β globulin (non-selective glomerular) [14].

Urinary protein is made up of proteins that get filtered through the glomerulus of the kidney and proteins produced by the kidney and urinary tract. Some urinary proteins may be more stable than blood-based proteins, and thus their detection may be more sensitive in the urine than in the blood [14].

Metabolomics is the scientific study of chemical processes involving cell metabolism. Metabolites are composed of intermediates of gene and protein expression or through a metabolism process. Metabolomics has emerged as a method to discover metabolite biomarkers for detecting, diagnosing, and managing HCC [92]. The 1H-NMR is a technology that can be used for metabolomics study, to detect metabolite markers in the urine of patients with HCC [91].

MS-based technologies are one of the most powerful tools for analyzing the proteome. The most common approaches in metabolomics involve gas chromatography-mass spectrometry (GC-MS) [93], liquid chromatography-mass spectrometry (LC-MS) [93,94], or nuclear magnetic resonance spectroscopy (NMR) [95]. Many independent authors have utilized quantitative techniques, such as 1H nuclear magnetic resonance and mass spectrometry, to discover novel biomarkers to aid the early diagnosis of HCC [50].

In a discovery cohort of patients with HCC (*n* = 40), urinary AFP (uAFP) and orosomucoid 1 (u-ORM1) were discovered by using iTRAQ and mass spectrometry [88]. iTRAQ is a method used to detect proteins. In the study, the authors demonstrated that the AUC of the combined uAFP and uORM1 was 0.864, which was higher than the AUC for uAFP or u-ORM1 alone. Urinary point-of-care to detect uAFP + ORM1 combination in patients with HCC can accurately discriminate it from patients without HCC, and would be an ideal test for use in low-recourse settings [96].

In another study, the authors used a proteomic strategy to detect proteomic biomarkers in the urine of patients with HCC. The controls comprised patients with chronic hepatitis B and without HCC. The authors combined seven urinary proteomic biomarkers to develop an assay with a good performance (AUC of 0.92 in the training dataset, AUC of 0.87 in the testing cohort), to discriminate HCCs from non-HCCs [97].

Only one urinary proteomic study about HCC found that S100A9 and GRN (Granulin) were elevated in HCC more than in normal control samples [98]. Two mediators of inflammation were positively identified: S100A9 and granulin protein markers, which belong to the cytoplasmic alarmin family of the host innate immune system. These HCC-associated cancer-specific biomarkers may have contributing roles not only in the dysregulated processes associated with various inflammatory and autoimmune conditions, but also in the tumorigenesis and cancer metastasis [98]. Zhang (2013) concluded that glycocholic acid, a secondary bile acid, was identified in the urine of patients with HCC [83].

Urinary volatile organic compounds have also been explored as potential urine-based biomarkers in patients with HCC. In a pilot study, 58 study subjects, comprising 20 HCCs and 38 non-HCC controls, were recruited. The controls were made up of patients with hepatic fibrosis and non-fibrosis. Using gas chromatography-ion mobility spectrometry (GS-IMS), at the AUC of 0.97, GS-IMS was able to differentiate HCC cases from hepatic fibrotic cases, and at the AUC of 0.62, GS-IMS discriminated HCC from non-fibrotic cases [99].

Among the proteoantigens that may have a role as HCC biomarkers and can be detected in urine are osteopontin (OPN) and heat shock protein (HSP). OPN is a versatile protein and has been implicated in various signaling pathways, promoting cancer cell proliferation, angiogenesis and metastasis in multiple cancers, including HCC [100]. In HCC, plasma and liver cancer tissue expressions of OPN are highly upregulated [101] and have been demonstrated as potential biomarkers for HCC early detection [102]. OPN can be detected in urine, suggesting that urinary OPN may be a urinary biomarker for HCC detection [103]. Heat shock proteins are evolutionarily conserved proteins that are significantly expressed in response to stress conditions, including tumor development and progression [104]. HSPs are typically named based on their molecular weight (HSP1, HSP 6 or HSP 70 etc.). HSPs have been demonstrated to be differentially upregulated in HCCs, and thus implicated as potential biomarkers for HCC diagnosis and prognosis [104]. Abd El-Salam et al. demonstrated that both blood and urine mRNAs of the HSP60 were significantly higher in patients with HCC versus controls [105]. This suggests that urinary HSPs are potential biomarkers for HCC.

In summary, the above studies highlighted the potential and various technologies that can be used to develop urine-based biomarkers for HCC detection. From the patient’s perspective, urinary markers may be more appealing, as their collection is non-invasive. Future prospective studies are required to validate these markers before their clinical application can be recommended.

## 4. Conclusions

Considering the targeted burden of HCC in resource-limited settings of sub-Saharan Africa and Southeast Asia, a simple, affordable, and at-the-point-of-need diagnosis test is highly desirable and urgently needed. These test characteristics are likely easier to implement widely and effectively, especially in resource-deprived settings. Assays or methods, either utilizing a single marker or as a combination of several biomarkers, that have been successfully utilized in developed countries may not be suitable in resource-limited regions of the world, where their application is relatively expensive and often requires advanced laboratory and clinical capabilities for implementation. Of the circulating biomarkers, protein markers from a blood test, such as AFP and DCP, are currently the most used. They are useful for risk assessment. Several blood-based DNA biomarkers are under development; however, trained medical personnel would be required to administer blood-based biomarkers. The collection of urine-based biomarkers is non-invasive and may be more receptive from the patient’s perspective. If urine can be collected at home, shipped to a testing center at room temperature, and subsequently used for detection by a simple assay, with a promising performance to discriminate HCC from non-HCC, it thus would be an ideal test in low-resource settings.

To date, the use of clinically useful biomarkers for the detection and management of HCC is largely based on case-control studies or incomplete prospective studies, therefore judgment on their use would be needed until more prospective data become available. The development of clinically useful biomarker(s) in HCC has been an active area of investigation. This, coupled with improved detection technologies and optimization of biomarkers, shows the promise of delivering a clinically useful test for the early detection and management of HCC in resource-limited settings, and should be realized in the near future.

## Figures and Tables

**Figure 1 diagnostics-13-00676-f001:**
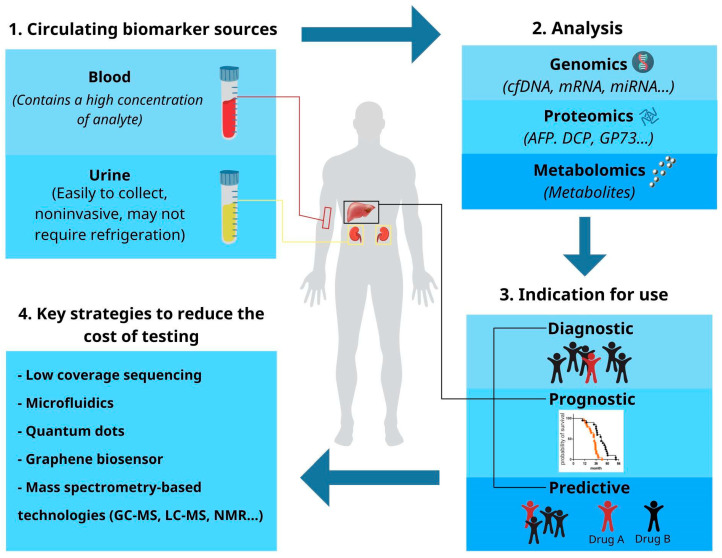
Circulating biomarkers of HCC. Shown are two types of circulating biomarkers (blood and urine, a focus of this review), the proposed strategies to reduce the cost of testing, and their potential applications in LRS for HCC screening, diagnosis, and management.

**Table 4 diagnostics-13-00676-t004:** Comparison of urine ctDNA test to other commonly used tests for HCC screening.

Test Status	Marker/Gene Panel	Sensitivity/Specificity
Most used	Serum AFP/L3-AFP	50/90%
Ready for use	Serum GP73	62–75%/63–97%
	Serum DCP	23–57%/70–90%
Recently developed	Urine ctDNA test for HCC	80/90%

## Data Availability

Not appliable.

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
