# Peer review of "Circulating Biomarkers for the Early Diagnosis and Management of Hepatocellular Carcinoma with Potential Application in Resource-Limited Settings"

_diagnostics, 2023, doi:10.3390/diagnostics13040676_

Round 1
Reviewer 1 Report
I would like to congratulate the authors of the manuscript entitled “Circulating Biomarkers for the Early Diagnosis and Management of Hepatocellular Carcinoma with Potential Application in Resource-Limited Settings” for conducting a very well-documented and review of literature, that sheds light upon a Hepatocellular Carcinoma related topic, regarding the potential applications in resource-limited settings of circulating biomarkers and their role in screening, diagnosis and prognosis for achieving early diagnosis and adequate management.
The manuscript’s topic is considered of actuality and interest, the authors present in a clear manner up-to-date information regarding current landscape of circulating HCC biomarkers. The current paper successfully takes into account the global spread of HCC burden, and thus focuses on biomarker applicability in the particular resource-limited settings.
The manuscript is well structured and very coherent, thus leading to a high quality of presentation.
The conclusions are pertinent and well supported by the content of the manuscript, additionally they outline key directions for future research. The cited references are considered up-to-date, and relevant to the research in question.
However, to appeal our most demanding readers I suggest improving the manuscript scientific wise, by addressing the following minor content issue:
The “Conclusion” section cannot contain bibliographic references. The manuscript’s conclusions are in strict relation to the author’s expertise in the field in question. This section should reveal the author’s opinion and contribution related to the studied topic.
Author Response
On behalf of my co-authors, I would like to thank you for the opportunity to revise and re-submit our manuscript- manuscript ID: diagnostics-2122388, titled, "Circulating Biomarkers for the Early Diagnosis and Management of Hepatocellular Carcinoma with Potential Application in Resource-Limited Settings". We found reviewer #1's comments to be constructive in revising the manuscript and have carefully responded to the feedback. Specifically, in the
Conclusion, we removed the bibliographic references and revised the section to reflect our expertise. We enclosed both the revised manuscript "with changes marked" in Simple Markup mode and the clean "unmarked" revised version.
Thank you again for your consideration of our revised manuscript. Please let us know if you need further information.
Sincerely,
Doan Dao, M.D.
On behalf of all authors
Reviewer 2 Report
This review is well-written and comprehensive of recent references. I coul suggest the authors insert only some citations about proteoantigens such as osteopontin, heat shock protein...
Author Response
On behalf of my co-authors, I would like to thank you for the opportunity to revise and re-submit our manuscript- manuscript ID: diagnostics-2122388, titled "Circulating Biomarkers for the Early Diagnosis and Management of Hepatocellular Carcinoma with Potential Application in Resource-Limited Settings". We found reviewer #2's suggestion to be constructive. We have added a paragraph (lines 686 to 700) describing proteo-antigens osteopontin and heat
shock protein and their potential roles as biomarkers in HCC. We enclosed both the revised manuscript "with changes marked" in Simple Markup mode and the clean "unmarked" revised version.
Thank you again for your consideration of our revised manuscript. Please let us know if you need further information.
Sincerely,
Doan Dao, M.D.
On behalf of all authors